# Rethinking Support Aggregation: Similarity-Based Weighting for In-context Medical Segmentation

**Kuan-I Chung**[1,2]                                   KUAN-I.CHUNG@VANDERBILT.EDU
[1] *Department of Computer Science, Vanderbilt University, TN USA*
[2] *Vanderbilt Institute for Surgery and Engineering, TN, USA*

**Daniel Moyer**[1,2]                                   DANIEL.MOYER@VANDERBILT.EDU

## Abstract

In-context medical segmentation models such as UniverSeg condition prediction on a support set of labeled examples, typically treating those examples equally. However, the relevance of support example to a given query can vary significantly. We propose a simple similarity-based weighting strategy that reweights support example's contributions without changing underlying model architecture. Experiments on CTSpine1K and WBC demonstrate improvement over uniform-weighed baseline.

**Keywords:** Segmentation, In-context Learning

## 1. Introduction

In-context medical image segmentation aims to adapt foundation segmentation model to new anatomical structure or imaging domain by leveraging a small number of annotated data. Recent works, e.g. UniverSeg (Butoi et al., 2023), achieved strong generalization with flexible task-conditioning on a support set of image-label pairs. However, UniverSeg treats all support examples as contributing equally, ignoring context example's relevance to the query image. Prior work has explored similarity-based selection or filtering of support examples (Gao et al., 2025), often relying on discrete selection and filtering algorithms. In this work, we assign similarity-based weights to support examples, allowing more relevant supports to contribute more during aggregation.

## 2. Method

The proposed method is built upon UniverSeg, an in-context medical segmentation model. We apply similarity-based weights at each cross block in UniverSeg for aggregating support features. This modification is applied at inference time and does not involve retraining.

**Similarity between the Query and the Supports:** To calculate the similarity, we leverage MobileSAM's (Zhang et al., 2023) image encoder $f$ which was distilled from original SAM (Kirillov et al., 2023) on natural images. First, we use this encoder to extract images' features. And then we calculate cosine similarity on these extracted features. Formally, the cosine similarity between the query image and the $k$-th support image is

$$\text{sim}_k = \frac{f(X_q^{(0)})^T f(X_{s_k}^{(0)})}{\left\| f(X_q^{(0)}) \right\|_2 \left\| f(X_{s_k}^{(0)}) \right\|_2},\tag{1}$$

where $f$ is the image encoder, $X_q^{(0)}$ is the query image, and $X_{s_k}^{(0)}$ is the $k$-th support image.

## 2.1. UniverSeg and Weighted Cross Block

UniverSeg aggregate support features via cross block by averaging intermediate representations across support samples (Butoi et al., 2023). In contrast, the proposed method assigns weights based on image similarity (see Figure 1). Formally,

$$X_q^{(t)} = \sum_{k=1}^{K} w_{s_k} X_{s_k}^{\prime(t)}, \text{ where } w_{s_k} = \frac{\exp(\frac{\text{sim}_k}{\tau})}{\sum_k \exp(\frac{\text{sim}_k}{\tau})}, \tag{2}$$

$X_{s_k}^{\prime(t)}$'s are intermediate support features from $\text{conv}_1$. Temperature $\tau > 0$ controls the sharpness of weighting: small $\tau$ tends to one-hot selection; large $\tau$, uniform weighting.

## 3. Experiments

### 3.1. Model and Data

We use the pretrained weights released by UniverSeg authors for testing our weighting strategy. We evaluate on CTSpine1K and WBC, both unseen during training. CTSpine1K is reduced to 2D spine segmentation, and WBC includes nucleus and cytoplasm tasks.

### 3.2. Support Size, Temperature, and Leave-one-out Testing

We evaluate three support set size:

$$K_{\text{full}} = |\mathcal{D}| - 1, \text{ where } |\mathcal{D}| \text{ is the size of dataset} \tag{3}$$

$$K_{\text{large}} = 2^{\lfloor \log_2 K_{\text{full}} \rfloor} \tag{4}$$

$$K_{\text{small}} = 2^{\lfloor \log_2 K_{\text{full}} \rfloor - 1} \tag{5}$$

We also test different temperature $\tau$ from $10^{-5}$ to $100$. Each time we pick one image as query and leave the rest as support set candidates. We compared the proposed method with the original UniverSeg.

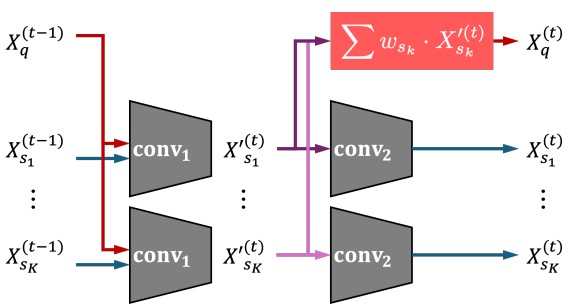

Figure 1: Weighted cross block with similarity-based aggregation.

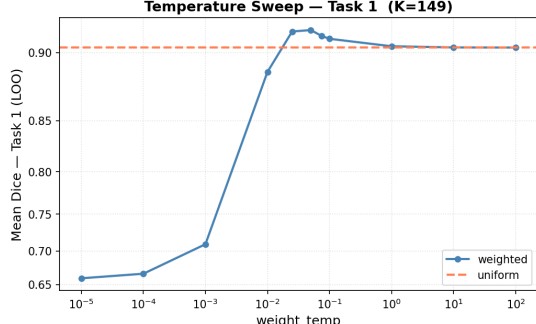

Figure 2: Temperature sweep with $K = 149$ on CTSPine1K.

Table 1: Leave-one-out Mean Dice (%). For each dataset, $K_{\text{full}} = |\mathcal{D}| - 1$. Smaller support sizes, $K_{\text{large}}$ and $K_{\text{small}}$, are selected on a logarithmic scale according to Equations (4) and (5). Since we randomly choose $K_{\text{larger}}$ and $K_{\text{small}}$ samples from $K_{\text{full}}$, we report the average Dice scores and their standard deviation from three seeds.

| Dataset($K_{\text{full}}$) | CTSpine1K (149) | WBC Dataset 1 (299) | | WBC Dataset 2 (99) | |
| Task | Spine | Nuclei | Cytoplasm | Nuclei | Cytoplasm |
|---|---|---|---|---|---|
| uniform $- K_{\text{full}}$ | 90.30 | 95.30 | 93.07 | 95.36 | 87.51 |
| ours | $\tau = 0.05$ | $\tau = 0.025$ | $\tau = 0.025$ | $\tau = 0.05$ | $\tau = 0.05$ |
| $K_{\text{small}}$ | 90.17±.17 | 95.22±.09 | 93.69±.15 | 94.66±.69 | 85.71±.87 |
| $K_{\text{large}}$ | 91.21±.07 | 95.64±.02 | 94.10±.01 | 95.56±.20 | 88.21±.59 |
| $K_{\text{full}}$ | **91.30** | **95.69** | **94.15** | **95.75** | **89.16** |

### 3.3. Results

In Figure 2, we demonstrate the weighting strategy's performance changes across different temperature $\tau$. Far left and right tend towards one single support image and uniform weighting respectively, while an intermediate temperature benefits the Dice score improvement. This phenomenon is expected from Equation (2). In Table 1, we show our strategy with $\tau$'s obtaining the best performance in each dataset tested. This result shows that using the full support set and our method outperforms the uniform baseline. In addition to improving performance at full support size, our weighting strategy allows for a reduced support size without losing competitive performance. When using less than half of support set ($K_{\text{small}}$), similarity-based weighting strategy can maintain performance close to full-support baseline.

## 4. Discussion

Our results suggest that not all support examples contribute equally in in-context segmentation. By assigning higher weights to more relevant samples, the proposed method reduces the influence of less informative support. This leads to consistent improvement at full-size support set, while maintaining competitive performance on smaller size of support set.

This method is simple, training-free, and integrates directly with existing architecture. It can be interpreted as introducing a form of cross-attention at the image level, where support contributions are modulated by similarity. However, because the cross block aggregation and similarity operate at image level, it may not capture task-specific or spatial localized relevance. A future extension is to model attention-based patch-level relevance.

### Acknowledgments

Research reported in this publication was supported by the Advanced Research Projects Agency for Health (ARPA-H) under the ALISS project, Award Number D24AC00415-00. The ARPA-H award of up to $11,935,038 provided 100% of the financial support for this work. The opinions and findings in this paper are solely the responsibility of the authors and do not necessarily represent the official views of ARPA-H.

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

## Appendix A.

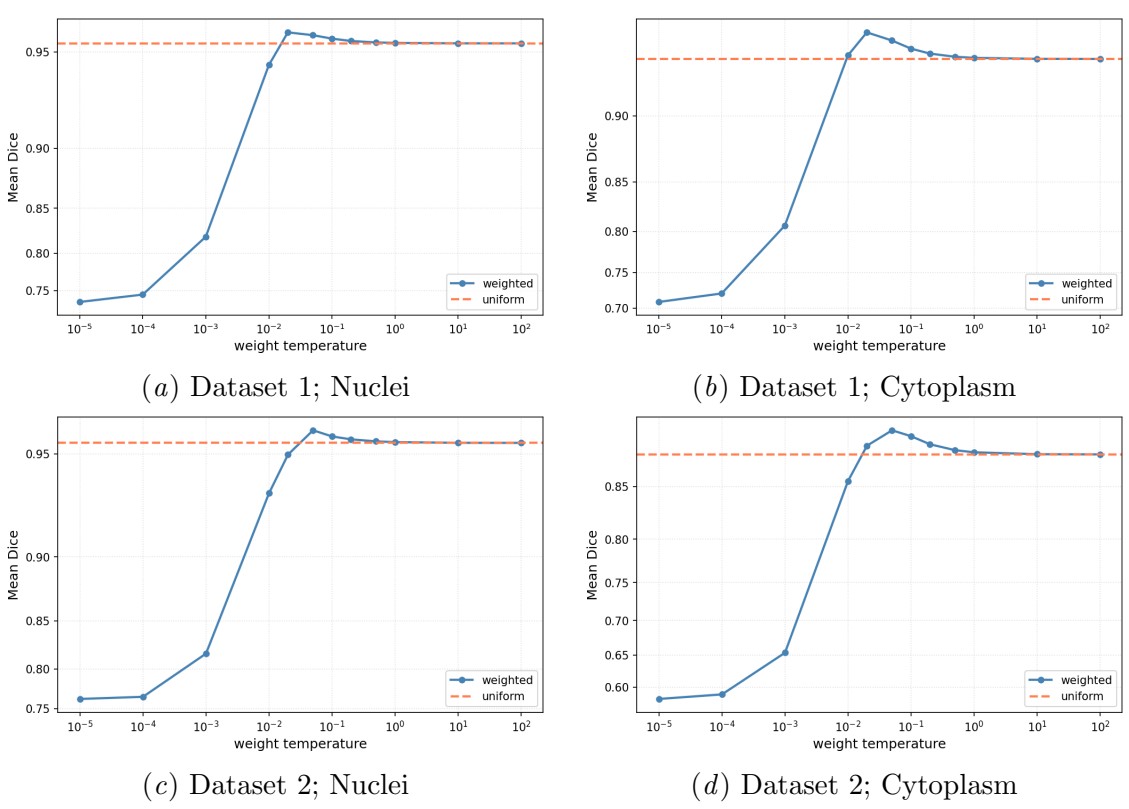

(*a*) Dataset 1; Nuclei        (*b*) Dataset 1; Cytoplasm

(*c*) Dataset 2; Nuclei        (*d*) Dataset 2; Cytoplasm

Figure 3: Sweeping temperature $\tau$ on WBC dataset

