# OpenReview forum: "Rethinking Support Aggregation: Similarity-Based Weighting for In-context Medical Segmentation"
_MIDL.io/2026/Short_Papers — MIDL 2026 - Short Papers Poster_

### Official Review · Reviewer_AxqD · 2026-04-29
**Weighting strategy for in-context segmentation**

**Rating:** 4
**Confidence:** 5

**Review:**

The paper is well written and structured. The experiments, conducted with three seeds and reporting standard deviation, demonstrate an improvement of the weighting strategy over the baseline. While the authors mention a brief direction for future work, they do not state the limitations of their method.

**Summary:**

The authors propose a similarity-based weighting for in-context medical segmentation to adapt foundation segmentation models models. Experiments on CTSpine1K (2D spine segmentation) and WBC (nucleus and cytoplasm tasks) demonstrate improvement over uniform-weighed baseline using UniverSeg as backbone.

**Strengths:**

- The authors report the average Dice scores and their standard deviation from three seeds.
- There is a brief mention to future work, "a future extension to model path-level relevance, similar to token-wise interaction in vision transformers".

**Weaknesses:**

- Introduction is quite brief and the related work is not comprehensive. However, given the short format, this is understandable.
- The authors do not justify the choice of the datasets (CTSpine1K and WBC).
- Table 1 could be better presented. The naming "uniform - Kfull" is confusing, as well as having the temperature within the table together with the Mean Dice (%).
- The authors do not discuss limitation of their proposed method.

**Justification Of Rating:**

The authors present a weighting strategy for in-context medical segmentation. They demonstrate improvement in performance on two datasets. I recommend accepting this work.

---

### Decision · Program_Chairs · 2026-05-08

Accept (Poster)